# Carbapenem-resistant Enterobacterales among patients with bloodstream infections in South Africa: Consolidated surveillance data, 2015–2021

Husna Ismail[1], Thembekile Buhle Christna Zwane[1,2], Elloise Du Toit[3],
Renata Maria Augusto da Costa[4]*, François Franceschi[4], Olga Perovic[1,5]*

1 Centre for Healthcare-Associated Infections, Antimicrobial Resistance and Mycoses at National Institute for Communicable Diseases, Division of the National Health Laboratory Service, Johannesburg, South Africa, 2 South African Field Epidemiology Training Programme at National Institute for Communicable Diseases, Division of the National Health Laboratory Service, Johannesburg, South Africa, 3 Global Antibiotic Research and Development Partnership, Cape Town, South Africa, 4 Global Antibiotic Research and Development Partnership, Geneva, Switzerland, 5 Department of Clinical Microbiology and Infectious Diseases, Faculty of Health Sciences, University of the Witwatersrand, Johannesburg, South Africa

* olgap@nicd.ac.za (OP); rdacosta@gardp.org (RMAdC)

## Abstract

### Background

A fifth of blood cultures from patients with infections acquired in a healthcare setting in South Africa will yield an organism. Carbapenem-resistant Enterobacterales (CREs), specifically *Klebsiella pneumoniae* are the most predominant Gram-negative bacteria (GNB) isolated among bloodstream infections (BSI). Additionally, the multidrug-resistant nature of these organisms is not only a threat to patients but it also poses a big public health challenge to current treatments and highlights the need for the development of new antimicrobials. Therefore, CRE have been placed on the critical priority list by the World Health Organization (WHO). We aimed to provide a holistic overview of the GERMS-SA CRE BSI surveillance data from 01 July 2015–31 December 2021.

### Methods

We conducted a cross-sectional study. CRE BSI was defined as the isolation of Enterobacterales resistant to any carbapenem (imipenem, ertapenem, meropenem and doripenem). All culture-confirmed cases captured on the database from 01 July 2015–31 December 2021 were considered for descriptive analysis and all cases with additional clinical information from 01 July 2015–31 December 2020 collected through the case report form (CRF) were used to perform analytical inferential statistics. We calculated the case fatality ratio of all cases with the outcome reported.

**Data availability statement:** All relevant data are within the paper and its Supporting Information files.

**Funding:** The author(s) received no specific funding for this work.

**Competing interests:** I have read the journal's policy and the authors of this manuscript and declare no competing interests.

## Results

Of the 5,258 culture-confirmed cases in the database for the surveillance period, the median age was 31 years (interquartile range 11–50) and 53.0% (2,787/5,258) were males. The majority of cases (64.6%, 1,361/2,017) were from Gauteng province that are participating in GERMS-SA surveillance. Of the cultures that isolated organisms, 53.0% (2,699/5,258) were sent to a reference laboratory for further testing. Case report forms were collected from 2,935 cases and these included audit cases. The clinical outcome was recorded for 99.5% (2,919/2,935) of the cases with a case fatality ratio of 36.1% (1,055/2,919). About 44.5% (1,298/2,916) of the cases had pre-existing conditions, most of which were other (n = 601). Most cases (78.4%, 2,288/2,920) received antibiotics on the current admission and 85.4% (2,488/2,912) had devices inserted. Of the 2,699 viable isolates, 2,100 (78%) were confirmed to be resistant to any of the carbapenems and 93.0% (n = 1,951) were carbapenemase-producing CREs. The carbapenem-resistant rate of all BSI was 1.9% (5,258/270,239). Of the CRE isolates, most were highly susceptible to fosfomycin (80.4%, 1,688/2,100). The potential significant risk factors for CRE BSI related mortality from multivariable logistic regression were being male, over 60 years, with pre-existing conditions, previously on antibiotics, mechanical ventilation as well as oxygenation, and previous hospital admission.

## Conclusion

Periodic surveillance for CRE should be performed on regular basis to support infection prevention and stewardship program at each facility.

## Introduction

Nosocomial infections, also known as healthcare-associated infections (HAIs), are caused by different types of organisms of which Gram-negative bacteria (GNB) are becoming increasingly prevalent as the causative agents for bloodstream infections (BSI) [1]. GNB can easily acquire antibiotic resistance, as seen with Enterobacterales, the largest group of bacteria which are commonly found in the human gastrointestinal tract [2,3]. *Klebsiella pneumoniae* is of particular concern as it accounts for a majority of organisms isolated from blood cultures and other specimens [3,4]. Enterobacterales have a high tendency of becoming multidrug-resistant (MDR) through various mechanisms including the acquisition of genes that confer resistance to different classes of antibiotics [3,5]. Carbapenems are the drug of choice for treating severe infections caused by MDR GNB. However, resistance to these carbapenems is becoming more prevalent worldwide and novel mechanisms of resistance are evolving rapidly [3,6,7]. Hence, the World Health Organization (WHO) has placed carbapenem-resistant Enterobacterales (CRE) on the critical priority pathogen list in need for research and development of new treatments [8].

According to the South African national laboratory-based surveillance, 20.0% of blood cultures tested yield an organism [9,10]. This surveillance programme focused on specific pathogens and their susceptibility to various antibiotics in order to monitor resistance patterns of critical pathogens. Through this surveillance, *K. pneumoniae* was found to be the most common organism isolated in hospitals (both public and private), and to be a significant contributor to the increase in carbapenem resistance over recent years (6.0–25.0%) [10]. Morbidity and mortality due to CRE infections are influenced by several factors both at the patient and healthcare level. At the patient-level, factors such as age and underlying medical conditions have been associated with worse patient outcomes [11,12]. From the healthcare side, admission to specific wards (specifically in intensive care), and past and current use of antibiotics are some of the factors associated with increased mortality in patients with CRE infections [11]. CRE infections have further downstream implications, with increased direct and indirect healthcare costs due to higher number of hospital admissions, longer hospital stays, and extended treatment regimens [13]. With the increase in resistance to carbapenems, there is a shortage of effective treatments.

Laboratory-based surveillance is a valuable tool for describing microorganisms and their corresponding antibiotic sensitivity. Supplementing clinical information with variable factors can support infection prevention and control (IPC) strategies in healthcare. The well-established surveillance platform GERMS-SA, conducts enhanced surveillance on selected hospitals, by collecting clinical information and carrying out confirmatory molecular testing on submitted isolates. Published reports from GERMS-SA have observed increased rates of resistance to carbapenems that correlated with the presence of carbapenemase genes in Enterobacterales isolates [11,14]. The expression of varied carbapenemases in clinical isolates results in a wider spectrum of hydrolytic activity against beta-lactam antibiotics, thus challenging the effectiveness of current and new combination treatments for CREs [15,16].

The CRE surveillance data collected by GERMS-SA previously published has included clinical characteristics of patients for the period 2015–2018, as well as factors associated with patient's mortality for their isolates collected from 2019 to 2020. However, it only included case patients above 18 months of age [11,14]. This report includes the consolidated epidemiology of CRE BSI cases collected from the start to the end of the surveillance period (01 July 2015–31 December 2021). In this study, we aimed to describe the main factors that were associated with mortality for all patients with available demographic and clinical information during the enhanced surveillance period, 01 July 2015–31 December 2020. We continued laboratory-based surveillance for another year, January to December 2021, but no patient's clinical characteristics were available.

## Methods

### Study design and setting

A cross-sectional study was conducted using surveillance data collected by the GERMS-SA surveillance platform from 14 enhanced sentinel hospitals (providing different levels of care) in four South African provinces namely, Gauteng, KwaZulu-Natal, Free State and Western Cape.

### Definitions

A CRE BSI case was defined as the isolation of Enterobacterales resistant to any carbapenem (imipenem, ertapenem, meropenem and doripenem). We applied a 21-day case definition to define a new case. Enterobacterales were regarded as the following genus and species: *Escherichia coli, Citrobacter* spp, *Enterobacter* spp, *Klebsiella* spp, *Morganella* spp, *Proteus* spp, *Providentia* spp, *Salmonella* spp and *Serratia* spp. An audit case was defined as a patient identified as meeting the surveillance case definition, whose information was captured on the laboratory information system but no isolate was sent to the National Institute of Communicable Diseases (NICD) for further testing.

## Collection of surveillance data

GERMS-SA sentinel hospitals linked to the National Health Laboratory Service (NHLS) network perform routine diagnostic testing and serve ~80% of the South African population [17]. The CRE surveillance system included enhanced surveillance, which was part of the GERMS-SA surveillance platform and defined a case as a patient whose blood culture grew an Enterobacterales organism that was non-susceptible (including both intermediate and resistant categories) to any of the carbapenems. These non-susceptible isolates have had their identity confirmed, and further phenotypic and genotypic characterization were carried out at a reference laboratory at the NICD. Furthermore, eligible patients would have demographic and clinical information collected by GERMS-SA surveillance officers at the enhanced sentinel hospitals. The patient and isolate data were entered onto a password protected Microsoft (MS) Access database. Audit cases from the NHLS laboratory information system (LIS) without an isolate sent to NICD were also entered onto this MS Access database. For this analysis, we extracted the data from MS Access and requested data from the NHLS corporate data warehouse (CDW) for all the clinical specimens that were collected and processed from the participating enhanced sentinel hospitals from 01 July 2015–31 December 2021. Data were accessed on 22 September 2022 and anonymised prior to analysis.

## Epidemiologic data analysis

Cases of CRE BSI including audit case data were used to describe the clinical epidemiology and microbiological characteristics. The clinical data were collected during the enhanced surveillance period (01 July 2015–31 December 2020) while the microbiological and demographic data covers the entire laboratory surveillance period (01 July 2015–31 December 2021).

## Phenotypic and molecular characterization of the isolates

All CRE BSI isolates that met the case definition were transported to the NICD. Viable CRE BSI isolates were characterized at the Centre for Healthcare-Associated Infections, Antimicrobial Resistance and Mycoses (CHARM), NICD. Confirmation of organism identification, antimicrobial susceptibility testing (AST) and molecular characterisation were performed using previously published methodology [14]. AST was performed using the MicroScan Walkaway system (Siemens Healthcare Diagnostics Inc., USA) with the NM44 card (Beckman Coulter Inc., USA), but colistin-resistant isolates were re-tested and confirmed with the Sensititre instrument (Trek Diagnostic Systems Ltd, UK) using the FRCOL panel (Separation Scientific SA (Pty) Ltd, SA). Results for the minimum inhibitory concentrations (MICs) were interpreted using the 2020 Clinical & Laboratory Standards Institute (CLSI) guidelines and the 2023 guidelines for colistin [18,19]. The breakpoints for tigecycline were interpreted using criteria from the Food and Drug Administration (FDA) [20]. For the molecular characterization of the CRE isolates, DNA was extracted from all CRE isolates using a crude boiling method and used to screen for the five most common carbapenemase genes ($bla_{GES}$ (GES-1–9 and 11), $bla_{IMP}$ (IMP-9, 16, 18, 22 and 25), $bla_{KPC}$, $bla_{NDM-1}$, $bla_{OXA-48-like}$ (i.e., OXA-162, 163, 181, 204, 232, 244, 245 and 247) and $bla_{VIM}$ (VIM-1–36) by a multiplex real-time polymerase chain reaction (PCR) assay (LightCycler 480 II; Roche Diagnostics Corp., USA) [21]. Colistin-resistant CRE isolates (with an MIC > 2 µg/mL) were screened for the presence of plasmid mediated colistin resistance (mcr)-1 to mcr-5 genes using a conventional multiplex PCR assay [22].

## Clinical outcome data analysis

Standardised case report forms (CRFs) were completed between 01 July 2015 and 31 December 2020 (enhanced surveillance period) and were included in this analysis. Descriptive statistics were used for the analysis of demographic and clinical data, particularly for cases that had completed CRF information. The case fatality ratio was determined as the proportion of cases with CRE that died over all the cases with known reported outcomes. Univariate and multivariable logistic

regression analysis were done to determine factors associated with death for cases with non-missing data on outcome and other variables. We accounted for the following health-care-related factors as confounders: invasive device insertion, current and past exposure to antibiotics, previous hospitalisation and mechanical ventilation (as a proxy for critically ill patients). Patient-related factors adjusted included underlying medical conditions (S1–S3 Tables). Additional model diagnostics were performed to consider variables in the final model (S1 Table). Predictive scoring systems (for example, SOFA or APACHE II score), which are useful for the prognostication of mortality in an intensive care unit (ICU) could not be calculated. Age and sex were included in the multivariable model regardless of p-values. Crude odds ratio (OR) and adjusted OR (aOR) were presented with 95% confidence intervals and p-values at the 0.05 significance level. All the analyses were performed using Stata version 15.1 (Stata Corp LP, College Station, Texas, USA).

### Ethical considerations

Approval to conduct this study was obtained from the Human Research Ethical Committee at the University of the Witwatersrand, ethical clearance number: M230985.

## Results

### Clinical epidemiology

A total of 2,428,894 laboratory specimens were collected and stored in NHLS CDW for GERMS-SA hospitals from 01 July 2015–31 December 2021. About 55.1% (1,339,343/ 2,428,894) were from blood cultures. Overall positivity of blood cultures was 20.1% (270,239/1,339,343). A total of 5,258 CRE BSI cases, as defined above, were entered into the GERMS-SA enhanced surveillance database. Forty-three percent (2,261/5,258) were audit cases with no isolate submitted to the NICD and 53.0% (2,699/5,258) were cases with viable isolates submitted for further testing (Fig 1).

### Patient demographics

Patient demographic data had some missing variables and as a result, the number of cases displayed in Table 1 are not the same as the total number of captured cases (n = 5,258). Males accounted for 53.6% (2,787/5,198) of the CRE BSI cases, while the median age was 31 years (interquartile range [IQR] 11–50). Sixty-one percent (3,237/4,869) of the CRE BSI cases were from the adult wards and the majority were from the Gauteng Province (68.1%, 3,588/5,258), the most populated province in the country.

### Microbiology characteristics

Further microbiological analysis was conducted on viable isolates received at CHARM at the NICD. A total of 2,699/5,258 viable isolates were confirmed as non-susceptible to any carbapenem (ertapenem, meropenem, imipenem and doripenem). About 77.4% (2,100/2,699) of the isolates, were confirmed to be resistant to any one of the carbapenems based on the CLSI criteria. The number of isolates submitted each year increased during the surveillance period. *Klebsiella spp* were the most prevalent among the CREs, of which *Klebsiella pneumoniae* accounted for 79.8% (1,676/2,100) followed by *Enterobacter cloacae* (6.2%, 132/2,100), *Serratia marcescens* (5.9%, 123/2,100) and *Escherichia coli* (2.3%, 49/2,100) (Fig 2).

### Antimicrobial susceptibility testing results

Of the 2,100 carbapenem resistant isolates, resistance to ertapenem was found in 97.2% of the isolates (2,043/2,100), followed by resistance to meropenem (70.7%, 1,485/2,100) and doripenem (67.2%, 1,412/2,100). Imipenem resistance was the least prevalent (55.4%, 1,163/2,100. All the CRE isolates showed resistance to multiple antibiotics (Fig 3). Beta–lactams (BL) and trimethoprim-sulfamethoxazole had resistance rates above 90% among the tested isolates. Penicillin-based

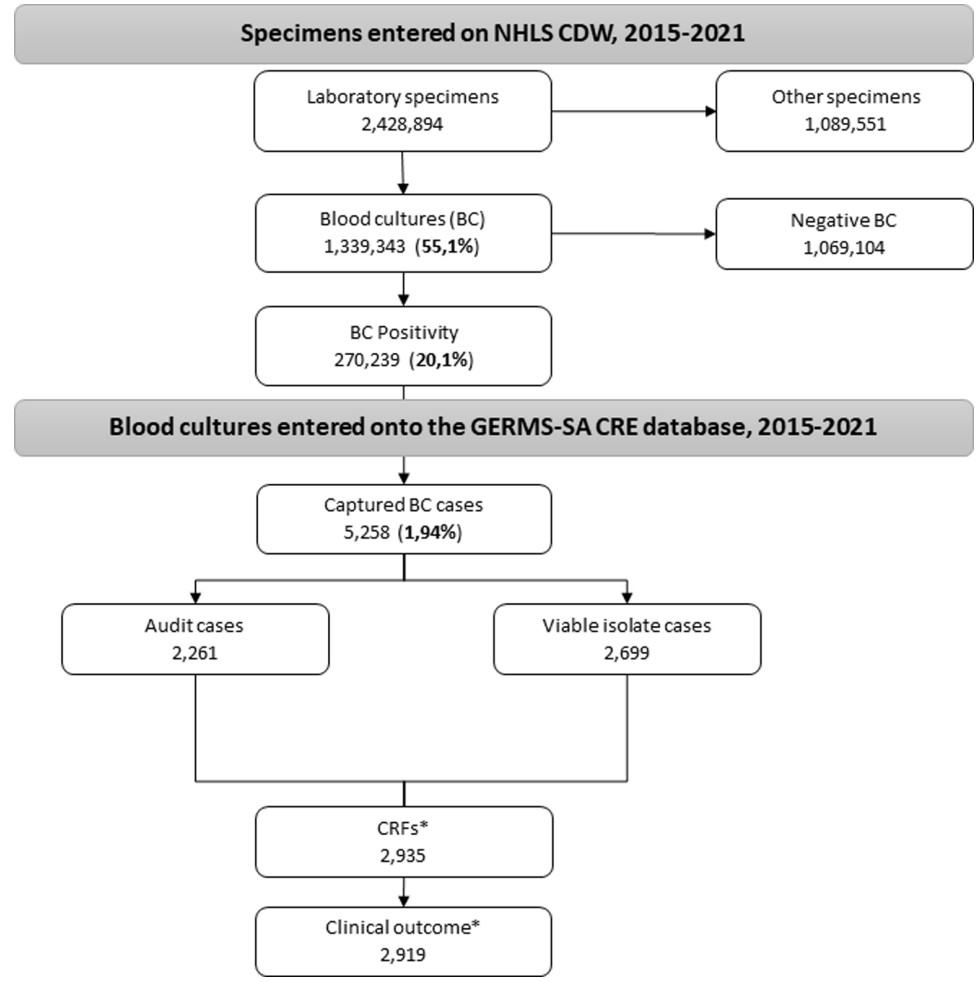

**Fig 1. Captured specimens on the NHLS CDW from the participating hospitals and cases reported on the GERMS-SA surveillance with completed CRFs.** NHLS- National Health Laboratory Service, CDW- corporate data warehouse, BC- blood culture, CRE- carbapenem-resistant Enterobacterales, CFR- case report forms. Other specimens included urine, respiratory, swabs and rectal specimens. *Case report forms with the clinical characteristics and outcome were collected during the enhanced surveillance, 01 July 2015 to 31 December 2021.

antibiotics with the highest resistance rates were ampicillin (100%, 2,100/2,100) and piperacillin (97.3%, 2,043/2,100). BL/Beta-lactamase inhibitor (BLI) combinations that exhibited the highest rates of resistance were amoxicillin-clavulanate (99.5%, 2,090/2,100) and ampicillin-sulbactam (99.4%, 2,087/2,100). Second, third and fourth generation cephalosporins displayed weak antibacterial activity, with resistance rates ranging from 98.4%, (2,066/2,100) for cefuroxime to 91.8%, (1,822/1,991) to cefepime. Lastly, 92.5% (1,943/2,100) isolates were resistant to trimethoprim-sulfamethoxazole. Susceptibility to other classes of antibiotics was high, ranging from 93.0% (1,952/2,100) for tigecycline to 80.4% (1,688/2,100) to fosfomycin, 62.6% (1,315/2,100) to amikacin, and 59% (1,239/2,100) to minocycline. Most of the isolates tested against colistin showed intermediate susceptibility (77.8%, 1,633/2,091).

## Molecular characterisation

The presence of carbapenemase genes was confirmed by PCR. Of the 2,100 CRE isolates tested, 93.0% (1,951/2,100) were carbapenemase-producing Enterobacterales (CPE) (Fig 4). Over the 7-year period, the proportion of $bla_{\text{OXA-48 \& variants}}$

**Table 1.** Demographic information of all CRE BSI cases (audits and viable isolates) admitted in GERMS-SA sentinel hospitals, 01 July 2015 to 31 December 2021.

| Demographic information all cases (audit cases and cases with isolates) | n* (%) |
|---|---|
| **Sex** | **5,198** |
| Female | 2,414 (46.4) |
| Male | 2,787 (53.6) |
| **Age category** | **4,987** |
| <30 days | 688 (13.8) |
| 1-11 months | 519 (10.4) |
| 1-9 years | 346 (6.9) |
| 10-19 years | 228 (4.6) |
| 20-29 | 446 (8.9) |
| 30-39 | 689 (13.8) |
| 40-49 | 617 (13.4) |
| 50-59 | 654 (13.1) |
| >=60 years | 800 (16.0) |
| **Ward type** | **4,869** |
| Paediatrics | 1,632 (33.5) |
| Adults | 3,237 (66.5) |
| **Province** | **5,258** |
| Gauteng | 3,588 (68.1) |
| Western Cape | 609 (11.6) |
| KwaZulu-Natal | 927 (17.6) |
| Free State | 134 (2.5) |

*Known data shown only, % column percentage.

increased and the proportion of $bla_{NDM}$ carbapenemases decreased. Additionally, 143 isolates co-harboured $bla_{OXA-48 \text{ & variants}}$ and $bla_{NDM}$ and 8 co-harboured $bla_{NDM}$ and $bla_{VIM}$ among others ([Fig 4]). Of the carbapenemase detected genes, $bla_{OXA-48 \text{ and the variants}}$ (75.0%, 1180/1584) were predominant in Gauteng, Free State and Western Cape provinces, while $bla_{NDM}$ (60.0%, 263/438) was predominant in KwaZulu-Natal ([Fig 5]).

## Clinical characteristics

During the enhanced surveillance period (01 July 2015–31 December 2020), 2,935 CRFs were collected from patients who met the case definition ([Table 2]). Clinical outcome was recorded for 99.5% (2,919/2,935) patients and reported an in-hospital case fatality ratio of 36.1% (1,055/2,919). Although most patients had clinical outcome information, other clinical characteristics in the CRF were missing, resulting in differing denominators below. Of the CRE BSI cases, 53.2% (1,298/2,439) had pre-existing conditions, and devices were inserted in 97.3% (2,488/2,558) of these cases. Most of the cases (89.2%, 2,288/2,564) received antibiotics on the current admission, and 37.8% (1,104/2,301) had past exposure (in the last 6 months) to antibiotics, 18.7% (448/2,396) were previously admitted and 33.2% (851/2,563) were mechanically ventilated.

After adjusting for factors that are likely associated with mortality among those with complete information showed that being male, over 60 years, with pre-existing conditions, previously on antibiotics, mechanical ventilation as well as oxygenation, and previous hospital admission were potential significant risk factors for CRE BSI mortality ([Table 3]).

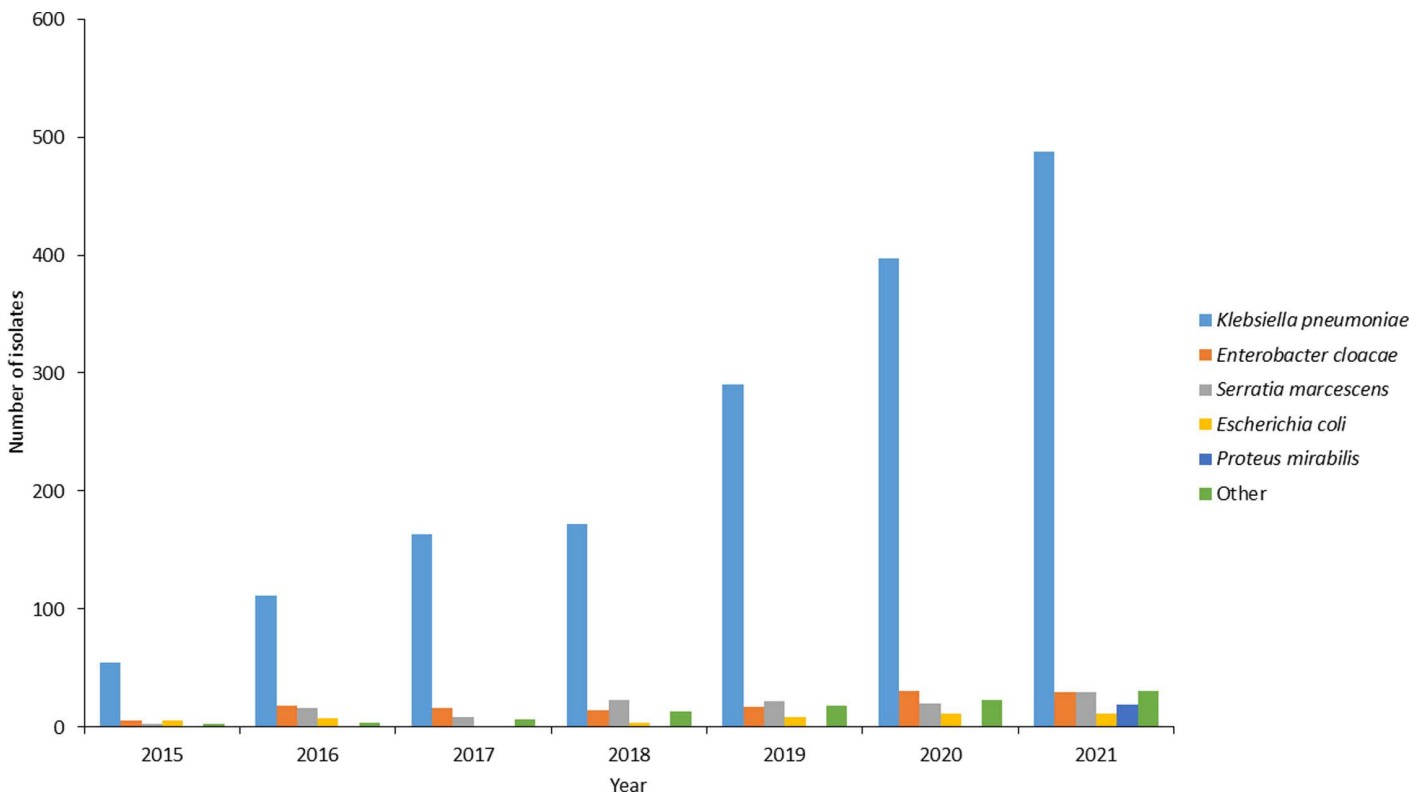

**Fig 2. Prevalent organisms isolated from patients with blood culture-confirmed CRE at GERMS-SA sentinel hospitals during the laboratory surveillance period from 01 July 2015 to 31 December 2021, n = 2,100.** Other = *Klebsiella varricola*, *Klebsiella oxytoca*, *Klebsiella aerogenes*, *Klebsiella ozaenae*, *Enterobacter kobei*, *Enterobacter asburiae*, *Citrobacter freundii*, *Citrobacter sedlakii*, *Providencia rettgeri*, *Providencia stuartti*, *Proteus vulgaris*, *Morganella morganii*, non-Typhi *Salmonella* and *Leclercia adecarboxylata*.

## Discussion

This analysis describes the CRE surveillance conducted in South Africa over the 7-year period and to augment previous reports with additional epidemiological data to investigate clinical correlations of disease severity and outcomes.

*Klebsiella pneumoniae* remained the predominant CRE organism in South Africa during the period of surveillance. This is in line with previous publications [11,14]. Furthermore, it is important to note that over the 7-year surveillance period, there was increase in the number of CRE organisms submitted to the NICD. This may suggest improved monitoring systems at these enhanced surveillance sites across South Africa or may be due to the development of resistance over time.

Our study was designed to select CREs from surveillance sites and we showed that 5,258/270,239 (1.94%) were resistant to carbapenems. Among the 2,699 submitted isolates, the susceptibility of Enterobacterales to imipenem, meropenem and doripenem ranged from 43.0%−54.0% while 75% of all the isolates tested were resistant to ertapenem. These findings were consistent with previously published surveillance data for different periods in South Africa and other studies in the Asia-pacific region [14,23,24]. Overall, our CRE isolates were susceptible to only a few antibiotics, 78.6% and 80.4% of the CRE isolates were susceptible to tigecycline and fosfomycin, respectively. Multidrug resistance among CRE isolates

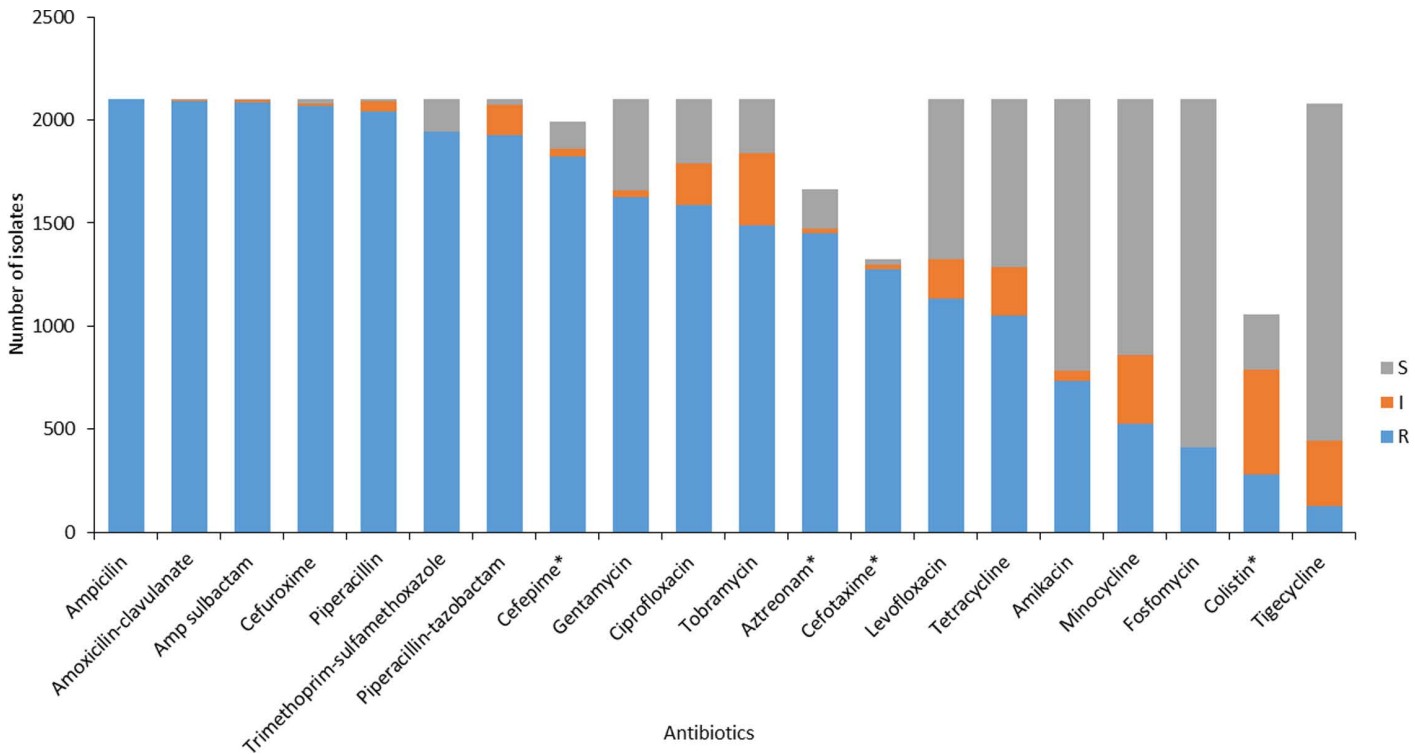

**Fig 3. Antibiogram of the confirmed CRE isolates at GERMS-SA sentinel hospitals from 01 July 2015 to 31 December 2021 (n = 2,100).** * Isolates tested for cefepime, aztreonam, cefotaxime and colistin had minimum inhibitory concentration values that were non-applicable (N/A). Therefore, the total number for those antibiotics is < 2,100. S-sensitive, I- intermediate and R-resistant. Tigecycline was interpreted using the FDA criteria [17].

was mostly observed against all generations of BL, including old BL-BLI combinations and trimethoprim-sulfamethoxazole with rates of resistance above 90%. A study conducted in 2019 in a Johannesburg hospital also reported that MDR isolates were resistant to penicillins and cephalosporins [25]. This is not surprising, given that penicillins and cephalosporins are all BL agents [26]. More than 75% of our isolates had intermediate susceptibility to colistin based on the CLSI guidelines for MIC method [16]. While this may indicate that colistin shows activity against CREs, colistin treatment is dose dependent [27].

Most of the CRE isolates (93%) harboured carbapenemase genes, in agreement with the global trend of carbapenem-resistance [28]. The carbapenemase-negative results seen in less than 10.0% of the isolates may be indicative of other mechanisms of resistance such as porin mutation, or the presence of a carbapenemase not tested for in the PCR panel. The $bla_{NDM}$ gene was one of the first and most prevalent carbapenemases detected in South Africa [25,29]. The present study confirms that in 2015, the presence of NDM was detected in 53% of the isolates. However, its prevalence has reduced along the years, being replaced by OXA-48-like carbapenemases (52.9%−83.9%) in all provinces. Interestingly, NDM was still the most prevalent carbapenemase in the KwaZulu-Natal province. This rapid increase in OXA-48 and variants has been reported globally [29,30]. In our study, a small percentage of CRE isolates (7.3%) harboured more than one carbapenemase gene, of which the most predominant were OXA-48, NDM and VIM. This co-harbouring of carbapenemase genes is due to the presence of multiple plasmids or other mobile genetic elements [31]. This further complicates the treatment of infections caused by CRE [15,16].

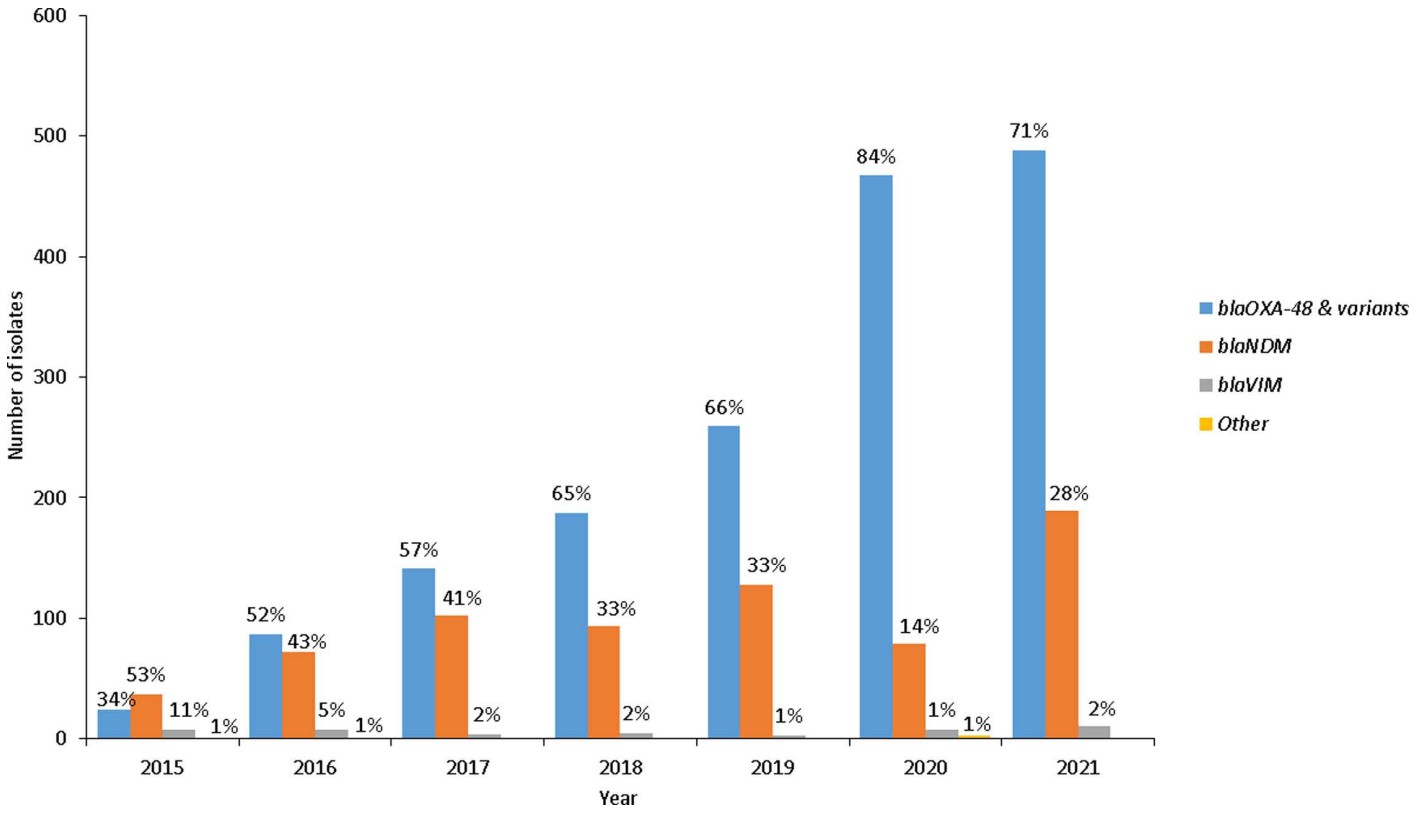

**Fig 4. Number of carbapenemase genes detected from blood culture-confirmed CPE isolates each year at GERMS-SA sentinel hospitals, 01 July 2015 to 31 December 2021.** Total number of isolates tested: 2015 (n = 70), 2016 (n = 180), 2017 (n = 248), 2018 (n = 287), 2019 (n = 390), 2020 (n = 557) and 2021 (n = 688). Other included isolates with low numbers: $bla_{KPC}$ (n = 3) in 2020 and $bla_{GES}$ (n = 1) in all the years except 2019.

Factors associated with mortality among CRE BSI cases that had CRFs completed. In our study older age (above 50 years) was associated with higher odds of death, which was similar to a study in China that found a high mortality caused by *K. pneumoniae* BSI in elderly patients in ICUs [12]. Our patients have pre-existing conditions resulting in more severe disease manifestation with unfavourable outcome, medical device insertion was not significantly associated with mortality, and this was the case in previous analyses of the same surveillance data (2019–2020). The presence of indwelling medical devices especially catheters has been reported as a risk factor for infection with CRE in other study [32]. While another study reported that early removal of the catheter was protective against mortality. The same was observed in the multivariable analysis although this was not significant [33].

As expected, exposure to antibiotics on current admission showed a protective effect against mortality. Previous antibiotic exposure remained a risk factor even on multivariable analysis which aligns with previous studies [12,34,35]. Additionally, a study reported that the length of previous hospital stay may confound the relationship between antibiotic use and carbapenem resistance leading to an increased risk of acquiring drug-resistant bacterium [35].

The previous admission was not associated with mortality in our multivariable analysis when we considered other factors. This is in contrary to a study that showed previous hospitalization and admission to the ICU to be independent predictors of mortality in multivariable analysis [34]. Lastly, mechanical ventilation, mental status and oxygenation also were risk factors associated with higher odds of mortality, this is in agreement with multiple studies that have reported that

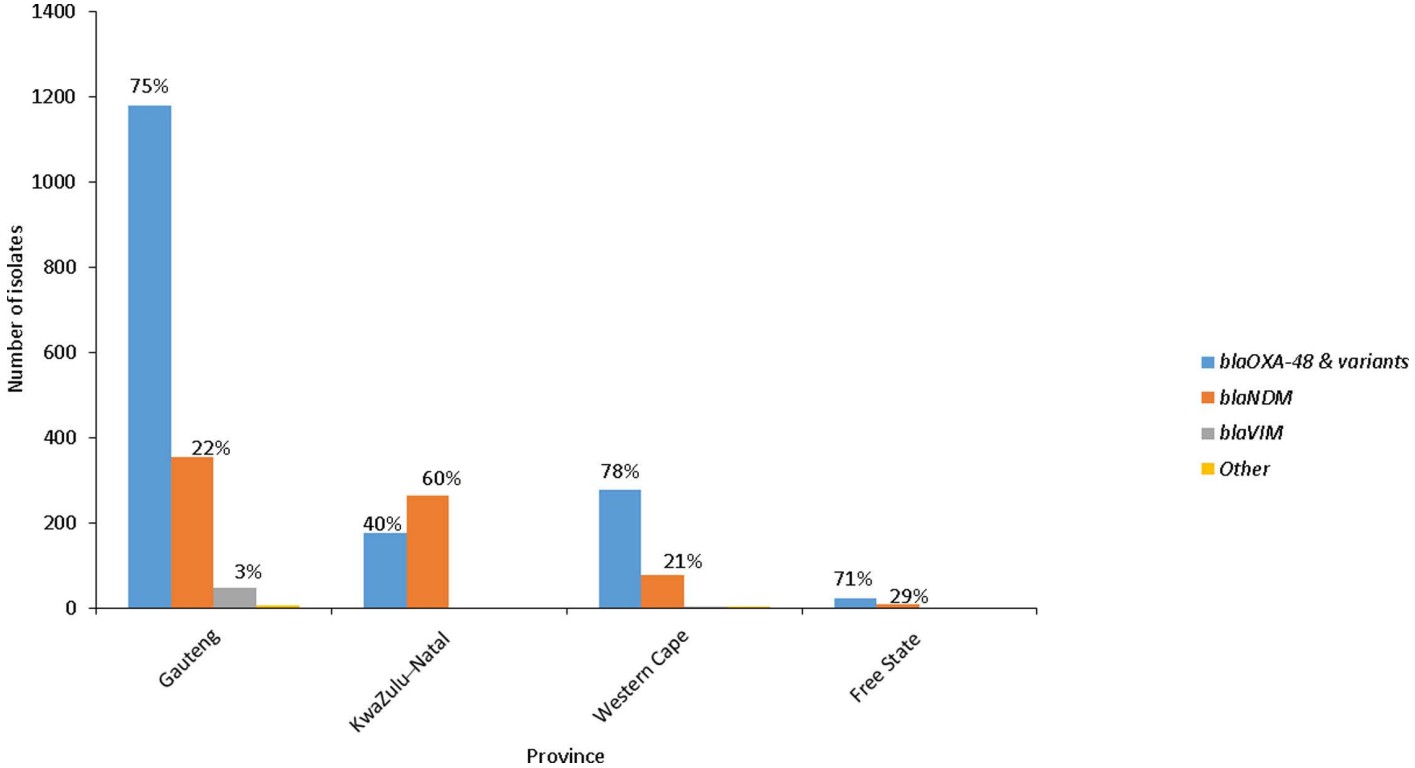

**Fig 5. Number of carbapenemase genes detected from blood culture-confirmed CRE isolates from the GERMS-SA sentinel hospitals in different provinces, 01 July 2015 to 31 December 2021.** Total number of isolates tested: Gauteng (n = 1584), KwaZulu-Natal (n = 438), Western Cape (n = 355) and Free State (n = 31). Other include number of isolated that detected $bla_{KPC}$ = 4 detected in Gauteng and $bla_{GES}$ detected in Gauteng (n = 1) and Western Cape (n = 2). Percentage of positive isolates presented at on top of the bars.

mechanical ventilation is a risk factor for CRE-associated BSI as well as mortality [36] These factors are generally considered for ICU admission and therefore may influence prognosis [37].

Adjusting for other known hospital-related and patient-level factors that may alter the association with the outcome is important [37]. The level of confounding or whether these factors are acting as possible effect modifiers needs further investigation, and the inclusion of a comparison group. In a study involving carbapenem-resistant GNB, the highest mortality was attributable to BSI caused by metallo-β-lactamase -producing CRE (35%) compared to that attributable to CRAB (16%).

Other cited studies specifically mention previous admission to ICU being associated with mortality; however, we could not establish patient admission beyond adult versus paediatric wards. In addition, ICU admission speaks to severe illness; therefore, it is important to have one of these two variables. The strength, however, remains that this study provides evidence of case fatality and factors that are associated with mortality among CRE BSI cases. Additionally, we analysed data for the entire surveillance period (01 July 2015–31 December 2021) which demonstrated an increasing trend in confirmed CPEs.

## Limitations

We acknowledge that there were several limitations with our inferential analysis. For instance, data on medical history was missing for a large proportion of cases. As a result, we could not determine and adjust for the severity of the disease.

**Table 2. Clinical characteristics of CRE BSI cases from the GERMS-SA sentinel hospitals with completed case report forms, 01 July 2015 to 31 December 2020.**

| Clinical Characteristics for cases with CRF | n* (%) |
|---|---|
| **Pre-existing condition** | **2,439** |
| No | 1,141 (46.8) |
| Yes | 1,298 (53.2) |
| **Antibiotic(s) received** | **2,564** |
| No | 276 (10.8) |
| Yes | 2,288 (89.2) |
| **Device inserted** | **2,558** |
| No | 70 (2.7) |
| Yes | 2,488 (97.3) |
| **Antibiotic(s) in past 6 months** | **2,301** |
| No | 1,197 (52.0) |
| Yes | 1,104 (48.0) |
| **Oxygenated** | **2,703** |
| No | 1,251 (46.3) |
| Yes | 1,452 (53.7) |
| **Mechanically ventilated** | **2,563** |
| No | 1,712 (66.8) |
| Yes | 851 (33.2) |
| **Previous hospital admission** | **2,396** |
| No | 1,921 (81.3) |
| Yes | 448 (18.7) |
| **Mental status** | **2,918** |
| Alert | 1,203 (41.0) |
| Disorientated | 236 (8.1) |
| Sedated | 761 (26.1) |
| Unconscious | 95 (3.2) |

*Known data shown only, % column percentage.

A major limitation was the laboratory-based retrospective nature of the study and the availability of clinical data for only a small set of patients, whose isolates could not be further characterised. Despite this limitation, our results are useful to better understand potential factors that may put patients at risk for carbapenem resistant infections and an adverse outcome. Most importantly, the rates of resistance to other agents such as colistin, are also increasing, heightening the urgency for availability of newer antibiotics that can tackle CRE infections effectively. This is a descriptive surveillance study over a period of 7 years, showing an increasing trend in number of CRE BSI and mortality associated to severity of diseases and previous use of antibiotics.

## Conclusion

A high number of Enterobacterales in this retrospective study highlighted the replacement of CPE genes as the proportion of $bla_{OXA-48 \& variants}$ increased and the $bla_{NDM}$ carbapenemases decreased. Clinical data showed that elderly male, with pre-existing conditions, previously on antibiotics, mechanical ventilation as well as oxygenation, and previous hospital admission were at significant risk factors for CRE BSI mortality. Structured surveillance program for CRE should be

**Table 3. Factors associated with mortality among CRE BSI cases with completed clinical information including outcome at GERMS sentinel hospitals, 01 July 2015 to 31 December 2020.**

| Characteristics | Alive n (row %) | Dead n (row %) | OR (95% CI) | p-value | aOR (95% CI) | p-value |
|---|---|---|---|---|---|---|
| **Sex** | | | | | | |
| Female | 739 (**58.4**) | 526 (**41.6**) | Ref | | Ref | |
| Male | 965 (**64.7**) | 525 (**35.2**) | 0.76 (0.65-0.89) | 0.001 | 0.80 (0.66-0.98) | 0.030 |
| **Age categories** | | | | | | |
| <30 days | 281 (**69.9**) | 121 (**30.1**) | 0.69 (0.52-0.93) | 0.015 | 0.77 (0.60-1.37) | 0.164 |
| 1-11 months | 201 (**72.8**) | 75 (**27.2**) | 0.60 (0.43-0.83) | 0.003 | 0.54 (0.36-0.82) | 0.004 |
| 1-9 years | 163 (**78.7**) | 44 (**21.3**) | 0.43 (0.30-0.64) | 0.000 | 0.37 (0.22-0.66) | 0.001 |
| 10-19 years | 94 (**70.1**) | 40 (**29.8**) | 0.68 (0.44-1.04) | 0.079 | 0.87 (0.51-1.48) | 0.621 |
| 20-29 | 166 (**66.4**) | 84 (**33.6**) | 0.81 (0.58-1.13) | 0.225 | 0.69 (0.44-1.07) | 0.098 |
| 30-39 | 243 (**61.7**) | 151 (**38.3**) | Ref | | Ref | |
| 40-49 | 182 (**56.0**) | 143 (**44.0**) | 1.26 (0.94-1.70) | 0.124 | 1.19 (0.81-1.75) | 0.374 |
| 50-59 | 189 (**54.8**) | 156 (**45.2**) | 1.32 (0.99-1.78) | 0.058 | 1.39 (0.95-2.04) | 0.086 |
| >=60 years | 196 (**45.2**) | 238 (**54.8**) | 1.95 (1.48-2.57) | <0.001 | 2.21 (1.54-3.16) | <0.001 |
| **Pre-existing conditions** | | | | | – | |
| No | 807 (**71.0**) | 329 (**28.9**) | Ref | | | |
| Yes | 759 (**58.6**) | 536 (**41.4**) | 1.73 (1.46-2.05) | <0.001 | | |
| **Medical device inserted** | | | | | – | |
| No | 48 (**68.6**) | 22 (**31.4**) | Ref | | | |
| Yes | 1,587 (**64.1**) | 889 (**35.9**) | 1.22 (0.73-2.03) | 0.442 | | |
| **Antibiotics on current admission** | | | | | | |
| No | 146 (**53.5**) | 127 (**46.5**) | Ref | | Ref | |
| Yes | 1,500 (**65.7**) | 783 (**34.3**) | 0.60 (0.46-0.77) | <0.001 | 0.59 (0.42-0.82) | 0.002 |
| **Previous exposure to antibiotics** | | | | | | |
| No | 854 (**71.9**) | 333 (**28.0**) | Ref | | Ref | |
| Yes | 655 (**59.8**) | 440 (**40.2**) | 1.72 (1.44-2.05) | <0.001 | 1.45 (1.18-1.79) | <0.001 |
| **Oxygenated** | | | | | | |
| No | 839 (**69.3**) | 372 (**30.7**) | Ref | | Ref | |
| Yes | 835 (**57.8**) | 609 (**42.2**) | 1.28 (1.18-1.39) | <0.001 | 1.67 (1.35-2.07) | <0.001 |
| **Mechanical ventilation** | | | | | | |
| No | 1,216 (**71.3**) | 489 (**28.7**) | Ref | | Ref | |
| Yes | 428 (**50.4**) | 421 (**49.6**) | 1.56 (1.43-1.70) | <0.001 | 2.76 (2.23-3.42) | <0.001 |
| **Previous hospital admission** | | | | | | |
| No | 1,295 (**67.8**) | 614 (**32.2**) | Ref | | Ref | |
| Yes | 266 (**59.8**) | 179 (**40.2**) | 1.19 (1.07-1.32) | 0.001 | 1.54 (1.18-1.79) | 0.002 |

performed on regular basis to support infection prevention and stewardship program at each facility. Public health actions such as development of guidelines and policies should utilize surveillance information for implementation of these measures and update them accordingly.

## Supporting information

**S1 Table. Initial model with all variables.**
(PDF)

**S2 Table. Bivariate analysis of ward type and pre-existing conditions.**
(PDF)

**S3 Table. Bivariate analysis of age categories with pre-existing conditions.**
(PDF)

**S4 Data. Carbapenem-resistant Enterobacterales dataset, 2015–2021.**
(PDF)

## Acknowledgments

This study data analysis was made possible by the Global Antibiotic Research and Development Partnership (GARDP) from the South African Medical Research Council (SAMRC). The authors acknowledge the GERMS-SA team for laboratory and epidemiology-based surveillance and the National Health Laboratory Service Corporate Data Warehouse (CDW) for the data.

## Author contributions

**Conceptualization:** Olga Perovic.

**Data curation:** Husna Ismail, Thembekile Buhle Christna Zwane, Olga Perovic.

**Formal analysis:** Husna Ismail, Thembekile Buhle Christna Zwane.

**Methodology:** Husna Ismail, Thembekile Buhle Christna Zwane.

**Resources:** Husna Ismail.

**Supervision:** Olga Perovic.

**Validation:** Husna Ismail, Olga Perovic.

**Visualization:** Husna Ismail, Thembekile Buhle Christna Zwane, Olga Perovic.

**Writing – original draft:** Husna Ismail, Thembekile Buhle Christna Zwane, Olga Perovic.

**Writing – review & editing:** Husna Ismail, Thembekile Buhle Christna Zwane, Elloise Du Toit, Renata Maria Augusto da Costa, François Franceschi, Olga Perovic.

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
