## [Decision Letter · Decision Letter 0]

Dear Dr. Perovic,

Thank you for submitting your manuscript to PLOS ONE. After careful consideration, we feel that it has merit but does not fully meet PLOS ONE’s publication criteria as it currently stands. Therefore, we invite you to submit a revised version of the manuscript that addresses the points raised during the review process.

We look forward to receiving your revised manuscript.

Kind regards,

Mohamed O Ahmed, Ph.D

Academic Editor

PLOS ONE

Journal Requirements:

“This study data analysis was made possible with funding received by the Global Antibiotic Research and Development Partnership (GARDP) from the South African Medical Research Council (SAMRC). The authors acknowledge the GERMS-SA team for laboratory and epidemiology-based surveillance and the National Health Laboratory Service Corporate Data Warehouse (CDW) for the data.”

3. We note that Figure S1 Fig in your submission contain [map/satellite] images which may be copyrighted. All PLOS content is published under the Creative Commons Attribution License (CC BY 4.0), which means that the manuscript, images, and Supporting Information files will be freely available online, and any third party is permitted to access, download, copy, distribute, and use these materials in any way, even commercially, with proper attribution. For these reasons, we cannot publish previously copyrighted maps or satellite images created using proprietary data, such as Google software (Google Maps, Street View, and Earth). For more information, see our copyright guidelines: http://journals.plos.org/plosone/s/licenses-and-copyright.

a. You may seek permission from the original copyright holder of Figure S1 Fig to publish the content specifically under the CC BY 4.0 license. 

Please upload the completed Content Permission Form or other proof of granted permissions as an 'Other' file with your submission.

Reviewers' comments:

Reviewer's Responses to Questions

**Comments to the Author**

1. Is the manuscript technically sound, and do the data support the conclusions?

Reviewer #1: No

Reviewer #2: Yes

2. Has the statistical analysis been performed appropriately and rigorously?

Reviewer #1: Yes

Reviewer #2: Yes

3. Have the authors made all data underlying the findings in their manuscript fully available?

Reviewer #1: Yes

Reviewer #2: Yes

4. Is the manuscript presented in an intelligible fashion and written in standard English?

Reviewer #1: Yes

Reviewer #2: Yes

Reviewer #1: This study was carried out with a sound methodology. It is well discussed in the manuscript.The conclusion is too general. However the conclusion is too general. The authors should rewrite the conclusion to specifically address and justify their findings.

Reviewer #2: Dear Authors, please review my comments and suggestions in the attached document. Additionally, ensure that abbreviations are used consistently and that comma delimiters are applied uniformly in the reporting of results.

**Do you want your identity to be public for this peer review?** For information about this choice, including consent withdrawal, please see our Privacy Policy

Reviewer #1: **Yes: ** Gülsüm İclal Bayhan

Reviewer #2: **Yes: ** Michelle Lowe

---

## [Author Response · Author response to Decision Letter 1]

29 Oct 2024

Dear Dr Emily Chenette,

Re: Carbapenem-resistant Enterobacterales among patients with bloodstream infections in South Africa: consolidated surveillance data, 2015-2021 – response to reviewers’ comments

Reviewer number 1

Reviewer1: The manuscript must describe a technically sound piece of scientific research with data that supports the conclusions. Experiments must have been conducted rigorously, with appropriate controls, replication, and sample sizes. The conclusions must be drawn appropriately based on the data presented – comment raised in the email body.

Authors’ response: Agreed, conclusion amended on lines 434-438 “A high number of Enterobacterales in this retrospective study highlighted the replacement of CPE genes as the proportion of blaOXA-48 & variants increased and the blaNDM carbapenemases decreased. Clinical data showed that elderly male, with pre-existing conditions, previously on antibiotics, mechanical ventilation as well as oxygenation, and previous hospital admission were at significant risk factors for CRE BSI mortality.”

Reviewer number 2

Reviewer2: Use manuscript abbreviations uniformly in the abstract or remove all abbreviations. Please double check the journal’s guidelines.

Authors’ response: Agreed, respective abbreviations were added.

Reviewer2: The abbreviation for Gram-negative bacteria was used in text. Please add it in the abstract as well (GNB).

Authors’ response: Agreed, (GNB) added on line 31.

Reviewer2: The abbreviation for World Health Organization was used in text. Please add it in the abstract as well (WHO).

Authors’ response: Agreed, (WHO) added on line 35.

Reviewer2: Please rephrase this sentence to enhance clarity.

Authors’ response: Agreed, sentence amended on line 35 “We aimed to provide a holistic overview of the GERMS-SA CRE BSI surveillance data.”

Reviewer2: Commas were used as delimiters for results in the text. Please standardize e.g. 1,361.

Authors’ response: Agreed, numbers were standardised using a comma as a delimiter on lines 44 to 55.

Reviewer2: The abbreviation for case report forms was used in text. Please add it in the abstract as well (CRF).

Authors’ response: Agreed, (CRF) added on line 42.

Reviewer2: Remove. The time period is mentioned in the M&M section.

Authors’ response: Agreed, sentence amended on line 48 “Case report forms were collected from 2,935 cases and these included audit cases”

Reviewer2: Standardise. Write out in full followed by the abbreviation.

Authors’ response: Agreed, (BSI) added on line 32. Sentence on lines 38-39 amended “CRE BSI was defined as the isolation of Enterobacterales resistant to any carbapenem (imipenem, ertapenem, meropenem and doripenem).”

Reviewer2: Standardise and add the n/n values.

Authors’ response: Agreed, sentence amended on line 54 “The carbapenem-resistant rate of all BSI was 1.9% (5,258/270,239).”

Reviewer2: Please add a reference to support this statement.

Authors’ response: Agreed, reference added on line 74.

Reviewer2: Please add a reference to support this statement.

Authors’ response: Agreed, reference added on line 83.

Reviewer2: Please clarify - what organisms? In the previous paragraphs only K. pneumoniae was discussed.

Authors’ response: Agreed, sentence amended on line 90 “Laboratory-based surveillance is a valuable tool for describing microorganisms and their corresponding antibiotic sensitivity.”

Reviewer2: Please define this abbreviation upon first use

Authors’ response: Agreed, sentence amended on lines 121-123 “An audit case was defined as a patient identified as meeting the surveillance case definition, whose information was captured on the laboratory information system but no isolate was sent to the National Institute of Communicable Diseases (NICD) for further testing.”

Reviewer2: Please add a reference to support this statement.

Authors’ response: Agreed, reference added on line 127.

Reviewer2: Check tenses

Authors’ response: Agreed, sentence amended on lines 130-132 “These non-susceptible isolates have had their identity confirmed, and further phenotypic and genotypic characterization were carried out at a reference laboratory at the NICD.”

Reviewer2: Please define this abbreviation upon first use

Authors’ response: already defined on line 135.

Reviewer2: Are you referring to CHARM? Please clarify.

Authors’ response: Yes. Sentence amended on line 149 “Viable CRE BSI isolates were further characterized at the Centre for Healthcare-Associated Infections, Antimicrobial Resistance and Mycoses (CHARM), NICD.”

Reviewer2: Please standardize comma delimiters. Please add “years” after the age groups

Authors’ response: Agreed, numbers were standardised using a comma as a delimiter on S1 Table1. Years was not analysed, therefore not included.

Reviewer2: Please standardize comma delimiters.

Authors’ response: Agreed, amended on line 205.

Reviewer2: Remove the full stop.

Authors’ response: Agreed, full stop removed on line 221.

Reviewer2: No footnote was added. Please add a footnote or alternatively remove the asterisk.

Authors’ response: Agreed, footnote added on line 224.

Reviewer2: Please see previous comment.

Authors’ response: Sentence amended on lines 227-228 “Further microbiological analysis was conducted on viable isolates received at CHARM at the NICD.”

Reviewer2: Add a comma to separate the percentage from the totals.

Authors’ response: Agreed, comma delimiter added on line 234.

Reviewer2: Standardise

Authors’ response: Agreed, comma delimiter added on line 244.

Reviewer2: Standardise

Authors’ response: Agreed, comma delimiter added on line 246.

Reviewer2: Standardise

Authors’ response: Agreed, comma delimiter added on line 246.

Reviewer2: Correct spelling.

Authors’ response: Agreed, spelling corrected for sulfonamide on line 247.

Reviewer2: Please correct.

Authors’ response: Corrected on line 257

Reviewer2: Standardise. Please check the rest of the document and correct.

Authors’ response: Agreed, comma delimiter added on line 262.

Reviewer2: Please correct the key legends - bla should be written in italics.

Authors’ response: Corrected on Fig 4A

Reviewer2: Please correct the key legends - bla should be written in italics.

Authors’ response: Corrected on Fig 4B

Reviewer2: Please correct “KwaZulu-Natal”

Authors’ response: Corrected in Fig 4B

Reviewer2: Check the use of brackets.

Authors’ response: Corrected on line 282.

Reviewer2: No footnote was added. Please add a footnote or alternatively remove the asterisk.

Authors’ response: Agreed, footnote added on line 303.

Reviewer2: Please add “years” after the age groups

Authors’ response: Years was not analysed, therefore not included.

Reviewer2: Add references.

Authors’ response: Agreed, references added on line 345.

Reviewer2: Add references.

Authors’ response: Agreed, references added on line 345, first part of our study findings.

Reviewer2: Class 2 carbapenems was not discussed or defined previously.

Authors’ response: Sentence amended on line 354 “Enterobacterales to imipenem, meropenem and doripenem ranged from 43.0%-54.0% while 75% of all the isolates tested were resistant to ertapenem”

Reviewer2: Please correct spelling.

Authors’ response: Spelling of sulphonamide is correct on line 360.

Reviewer2: MDR abbreviation was used previously. Please standardise the use of it.

Authors’ response: Agreed, MDR added on line 361.

Reviewer2: BL abbreviation was used previously. Please standardise the use of it.

Authors’ response: Agreed, BL added on line 361.

Reviewer2: Previously the beta-lactamase genes were defined as blaNDM etc. Please standardise terminology.

Authors’ response: Agreed, sentence amended on lines 370-371 “The blaNDM gene was one of the first and most prevalent carbapenemases detected in South Africa.”

Reviewer2: Please rephrase

Authors’ response: Agreed, sentence amended on lines 372-373 “However, its prevalence has reduced along the years, being replaced by OXA-48-like carbapenemases (52.9-83.9%) in all provinces”

Reviewer2: Not all references listed here support this statement. Please revise.

Authors’ response: Agreed, references added on line 375.

Reviewer2: CRF abbreviation was used previously. Please standardise the use of it.

Authors’ response: Agreed, CRFs added on line 381.

Reviewer2: “medical” - lower case “m”.

Authors’ response: Corrected on line 385.

Reviewer2: Mentioned studies but only one reference were given

Authors’ response: Agreed, references added on line 394.

Reviewer2: This abbreviation was not previously defined. Please write in full followed by the abbreviation.

Authors’ response: Agreed. MBL written out in full metallo-β-lactamase on line 409.

Reviewer2: Rephrase for better clarity. This doesn’t make sense.

Authors’ response: Agreed. Sentence amended on lines 421-424 “For instance, data on medical history was missing for a large proportion of cases. As a result we could not determine and adjust for the severity of the disease.”

Reviewer2: Please revise to specify CRE infections. If you are discussing carbapenem-resistant infections more

broadly, could you explain why you are using a more general term?

Authors’ response: Sentence amended on lines 426-427” Despite this limitation, our results are useful to better understand potential factors that may put patients at risk for infection with CRE and an adverse outcome.”

Reviewer2: Please rephrase to enhance clarity. These statements are not clear and a bit confusing.

Authors’ response: Agreed, conclusion amended on lines 434-438 “A high number of Enterobacterales in this retrospective study highlighted the replacement of CPE genes as the proportion of blaOXA-48 & variants increased and the blaNDM carbapenemases decreased. Clinical data showed that elderly male, with pre-existing conditions, previously on antibiotics, mechanical ventilation as well as oxygenation, and previous hospital admission were at significant risk factors for CRE BSI mortality.”

Reviewer2: Please add funding/grant reference numbers

Authors’ response: The journal requested that funding information be removed from the acknowledgment section

Reviewer2: Please refer to these tables in-text.

Authors’ response: Agreed, Sentence amended on lines 176-177 “Patient-related factors adjusted included underlying medical conditions (S1-3 Tables).”

Additional request by authors to partial financial disclosure: Could you, please, make this new statement in the acknowledgement?

“Epidemiological analysis was supported the DNDi GARDP Southern Africa (as a GARDP network member). The authors acknowledge the GERMS-SA team for laboratory and epidemiology-based surveillance and the National Health Laboratory Service Corporate Data Warehouse (CDW) for the data.” or add

---

## [Decision Letter · Decision Letter 1]

Dear Dr. Perovic,

Thank you for submitting your manuscript to PLOS ONE. After careful consideration, we feel that it has merit but does not fully meet PLOS ONE’s publication criteria as it currently stands. Therefore, we invite you to submit a revised version of the manuscript that addresses the points raised during the review process.

We look forward to receiving your revised manuscript.

Kind regards,

Mohamed O Ahmed, Ph.D

Academic Editor

PLOS ONE

Journal Requirements:

Reviewers' comments:

Reviewer's Responses to Questions

**Comments to the Author**

Reviewer #3: (No Response)

2. Is the manuscript technically sound, and do the data support the conclusions?

Reviewer #3: Yes

3. Has the statistical analysis been performed appropriately and rigorously?

Reviewer #3: Yes

4. Have the authors made all data underlying the findings in their manuscript fully available?

Reviewer #3: Yes

5. Is the manuscript presented in an intelligible fashion and written in standard English?

Reviewer #3: Yes

Reviewer #3: The manuscript presents a comprehensive surveillance study of carbapenem-resistant Enterobacterales (CRE) in South Africa from 2015 to 2021. The updated version demonstrates significant improvements addressing most of the reviewers' previous concerns. However, some additional issues should be addressed to further enhance the clarity, rigor, and overall flow of the manuscript.

Abstract and Clarity:

The modifications made to the abstract (lines 31–48) have improved its readability. Ensure that abbreviations (e.g., GNB, CRF) are used consistently across the entire abstract and that all abbreviations are clearly defined upon first use.

The sentence on lines 35–38 ("We aimed to provide a holistic overview...") is now clearer; however, further simplification would improve flow. Consider shortening by merging overlapping ideas.

Introduction:

The revised introduction effectively sets the context. However, some references added (lines 74 and 83) are quite general and may not fully support the specific claims. Ensure they are the most relevant sources.

Methods Section:

The additional clarifications in lines 121–132 improve the description of audit cases. However, the rationale for excluding certain samples could benefit from brief elaboration.

Lines 149–150 ("Viable CRE BSI isolates...") now reference CHARM clearly; this is a welcome improvement.

Results and Figures:

Figures 4A and 4B have been revised as requested, with corrections to italics for "bla" genes and corrected spelling for "KwaZulu-Natal." These changes enhance accuracy.

Check whether additional legends or annotations can further improve figure readability, particularly where percentages are presented alongside totals.

The overall presentation of statistical results is clearer. However, certain segments, such as the resistance trends (lines 354–373), are still slightly dense. Use bullet points where possible to improve readability.

Discussion:

The discussion was refined to address gaps in interpretation. However, ensure that references support each key assertion, especially regarding carbapenemase distribution.

Consider tightening the language in lines 421–427 and 434–438 to reduce redundancy and improve clarity.

Tables and Supplementary Materials:

The tables now include necessary details, but cross-referencing them within the text (e.g., S1–3 Tables) remains crucial for coherence. Double-check for proper placement of in-text citations.

Suggested Minor Edits:

Ensure uniformity in decimal formatting (e.g., percentages).

Confirm spelling consistency (e.g., "sulfonamide" vs. "sulphonamide").

Verify alignment between legends, figures, and main text descriptions.

**Do you want your identity to be public for this peer review?** For information about this choice, including consent withdrawal, please see our Privacy Policy

Reviewer #3: **Yes: ** Maurizio Sanguinetti

---

## [Author Response · Author response to Decision Letter 2]

14 Feb 2025

Ethics certificates that cover the whole period are provided

---

## [Decision Letter · Decision Letter 2]

Carbapenem-resistant Enterobacterales among patients with bloodstream infections in South Africa: consolidated surveillance data, 2015-2021

PONE-D-24-22805R2

Dear Dr. Perovic,

We’re pleased to inform you that your manuscript has been judged scientifically suitable for publication and will be formally accepted for publication once it meets all outstanding technical requirements.

Kind regards,

Mohamed O Ahmed, Ph.D

Academic Editor

PLOS ONE

---

## [Editor Report · Acceptance letter]

PONE-D-24-22805R2

PLOS ONE

Dear Dr. Perovic,

I'm pleased to inform you that your manuscript has been deemed suitable for publication in PLOS ONE. Congratulations! Your manuscript is now being handed over to our production team.

Kind regards,

on behalf of

Dr. Mohamed O Ahmed

Academic Editor

PLOS ONE